# Distributed estimation of the inverse Hessian by determinantal averaging

**Michał Dereziński**
Department of Statistics
University of California, Berkeley
mderezin@berkeley.edu

**Michael W. Mahoney**
ICSI and Department of Statistics
University of California, Berkeley
mmahoney@stat.berkeley.edu

## Abstract

In distributed optimization and distributed numerical linear algebra, we often encounter an *inversion bias*: if we want to compute a quantity that depends on the inverse of a sum of distributed matrices, then the sum of the inverses does not equal the inverse of the sum. An example of this occurs in distributed Newton's method, where we wish to compute (or implicitly work with) the inverse Hessian multiplied by the gradient. In this case, locally computed estimates are biased, and so taking a uniform average will not recover the correct solution. To address this, we propose *determinantal averaging*, a new approach for correcting the inversion bias. This approach involves reweighting the local estimates of the Newton's step proportionally to the determinant of the local Hessian estimate, and then averaging them together to obtain an improved global estimate. This method provides the first known distributed Newton step that is *asymptotically consistent*, i.e., it recovers the exact step in the limit as the number of distributed partitions grows to infinity. To show this, we develop new expectation identities and moment bounds for the determinant and adjugate of a random matrix. Determinantal averaging can be applied not only to Newton's method, but to computing any quantity that is a linear transformation of a matrix inverse, e.g., taking a trace of the inverse covariance matrix, which is used in data uncertainty quantification.

## 1 Introduction

Many problems in machine learning and optimization require that we produce an accurate estimate of a square matrix $\mathbf{H}$ (such as the Hessian of a loss function or a sample covariance), while having access to many copies of some unbiased estimator of $\mathbf{H}$, i.e., a random matrix $\widehat{\mathbf{H}}$ such that $\mathbb{E}[\widehat{\mathbf{H}}] = \mathbf{H}$. In these cases, taking a uniform average of those independent copies provides a natural strategy for boosting the estimation accuracy, essentially by making use of the law of large numbers: $\frac{1}{m}\sum_{t=1}^m \widehat{\mathbf{H}}_t \to \mathbf{H}$. For many other problems, however, we are more interested in the inverse (Hessian/covariance) matrix $\mathbf{H}^{-1}$, and it is necessary or desirable to work with $\widehat{\mathbf{H}}^{-1}$ as the estimator. Here, a naïve averaging approach has certain fundamental limitations (described in more detail below). The basic reason for this is that $\mathbb{E}[\widehat{\mathbf{H}}^{-1}] \neq \mathbf{H}^{-1}$, i.e., that there is what may be called an *inversion bias*.

In this paper, we propose a method to address this inversion bias challenge. The method uses a *weighted* average, where the weights are carefully chosen to compensate for and correct the bias. Our motivation comes from distributed Newton's method (explained shortly), where combining independent estimates of the inverse Hessian is desired, but our method is more generally applicable, and so we first state our key ideas in a more general context.

**Theorem 1** *Let $s_i$ be independent random variables and $\mathbf{Z}_i$ be fixed square rank-$1$ matrices. If $\widehat{\mathbf{H}} = \sum_i s_i \mathbf{Z}_i$ is invertible almost surely, then the inverse of the matrix $\mathbf{H} = \mathbb{E}[\widehat{\mathbf{H}}]$ can be expressed as:*

$$\mathbf{H}^{-1} = \frac{\mathbb{E}\big[\det(\widehat{\mathbf{H}})\widehat{\mathbf{H}}^{-1}\big]}{\mathbb{E}\big[\det(\widehat{\mathbf{H}})\big]}.$$

To demonstrate the implications of Theorem 1, suppose that our goal is to estimate $F(\mathbf{H}^{-1})$ for some linear function $F$. For example, in the case of Newton's method $F(\mathbf{H}^{-1}) = \mathbf{H}^{-1}\mathbf{g}$, where $\mathbf{g}$ is the gradient and $\mathbf{H}$ is the Hessian. Another example would be $F(\mathbf{H}^{-1}) = \mathrm{tr}(\mathbf{H}^{-1})$, where $\mathbf{H}$ is the covariance matrix of a dataset and $\mathrm{tr}(\cdot)$ is the matrix trace, which is useful for uncertainty quantification. For these and other cases, consider the following estimation of $F(\mathbf{H}^{-1})$, which takes an average of the individual estimates $F(\widehat{\mathbf{H}}_t^{-1})$, each weighted by the determinant of $\widehat{\mathbf{H}}_t$, i.e.,

$$\textbf{Determinantal Averaging:} \quad \hat{F}_m = \frac{\sum_{t=1}^m a_t F(\widehat{\mathbf{H}}_t^{-1})}{\sum_{t=1}^m a_t}, \qquad a_t = \det(\widehat{\mathbf{H}}_t).$$

By applying the law of large numbers (separately to the numerator and the denominator), Theorem 1 easily implies that if $\widehat{\mathbf{H}}_1, \ldots, \widehat{\mathbf{H}}_m$ are i.i.d. copies of $\widehat{\mathbf{H}}$ then this *determinantal averaging estimator* is asymptotically consistent, i.e., $\hat{F}_m \to F(\mathbf{H}^{-1})$, almost surely. This determinantal averaging estimator is particularly useful when problem constraints do not allow us to compute $F\big((\frac{1}{m}\sum_t \widehat{\mathbf{H}}_t)^{-1}\big)$, e.g., when the matrices are distributed and not easily combined.

To establish finite sample convergence guarantees for estimators obtained via determinantal averaging, we establish the following matrix concentration result. We state it separately since it is technically interesting and since its proof requires novel bounds for the higher moments of the determinant of a random matrix, which is likely to be of independent interest. Below and throughout the paper, $C$ denotes an absolute constant and "$\preceq$" is the Löwner order on positive semi-definite (psd) matrices.

**Theorem 2** *Let $\widehat{\mathbf{H}} = \frac{1}{k}\sum_{i=1}^n b_i \mathbf{Z}_i + \mathbf{B}$ and $\mathbf{H} = \mathbb{E}[\widehat{\mathbf{H}}]$, where $\mathbf{B}$ is a positive definite $d \times d$ matrix and $b_i$ are i.i.d. $\mathrm{Bernoulli}(\frac{k}{n})$. Moreover, assume that all $\mathbf{Z}_i$ are psd, $d \times d$ and rank-$1$. If $k \geq C\frac{\mu d^2}{\eta^2}\log^3\frac{d}{\delta}$ for $\eta \in (0,1)$ and $\mu = \max_i \|\mathbf{Z}_i\mathbf{H}^{-1}\|/d$, then*

$$\left(1 - \frac{\eta}{\sqrt{m}}\right) \cdot \mathbf{H}^{-1} \preceq \frac{\sum_{t=1}^m a_t \widehat{\mathbf{H}}_t^{-1}}{\sum_{t=1}^m a_t} \preceq \left(1 + \frac{\eta}{\sqrt{m}}\right) \cdot \mathbf{H}^{-1} \quad \text{with probability } \geq 1 - \delta,$$

*where $\widehat{\mathbf{H}}_1, \ldots, \widehat{\mathbf{H}}_m \overset{\text{i.i.d.}}{\sim} \widehat{\mathbf{H}}$ and $a_t = \det(\widehat{\mathbf{H}}_t)$.*

## 1.1 Distributed Newton's method

To illustrate how determinantal averaging can be useful in the context of distributed optimization, consider the task of batch minimization of a convex loss over vectors $\mathbf{w} \in \mathbb{R}^d$, defined as follows:

$$\mathcal{L}(\mathbf{w}) \overset{def}{=} \frac{1}{n}\sum_{i=1}^n \ell_i(\mathbf{w}^\top \mathbf{x}_i) + \frac{\lambda}{2}\|\mathbf{w}\|^2, \tag{1}$$

where $\lambda > 0$, and $\ell_i$ are convex, twice differentiable and smooth. Given a vector $\mathbf{w}$, Newton's method dictates that the correct way to move towards the optimum is to perform an update $\widetilde{\mathbf{w}} = \mathbf{w} - \mathbf{p}$, with $\mathbf{p} = \nabla^{-2}\mathcal{L}(\mathbf{w})\nabla\mathcal{L}(\mathbf{w})$, where $\nabla^{-2}\mathcal{L}(\mathbf{w}) = (\nabla^2\mathcal{L}(\mathbf{w}))^{-1}$ denotes the inverse Hessian of $\mathcal{L}$ at $\mathbf{w}$.[1] Here, the Hessian and gradient are:

$$\nabla^2\mathcal{L}(\mathbf{w}) = \frac{1}{n}\sum_i \ell_i''(\mathbf{w}^\top \mathbf{x}_i)\, \mathbf{x}_i\mathbf{x}_i^\top + \lambda\mathbf{I}, \quad \text{and} \quad \nabla\mathcal{L}(\mathbf{w}) = \frac{1}{n}\sum_i \ell_i'(\mathbf{w}^\top \mathbf{x}_i)\, \mathbf{x}_i + \lambda\mathbf{w}.$$

For our distributed Newton application, we study a distributed computational model, where a single machine has access to a subsampled version of $\mathcal{L}$ with sample size parameter $k \ll n$:

$$\widehat{\mathcal{L}}(\mathbf{w}) \overset{def}{=} \frac{1}{k}\sum_{i=1}^n b_i \ell_i(\mathbf{w}^\top \mathbf{x}_i) + \frac{\lambda}{2}\|\mathbf{w}\|^2, \quad \text{where} \quad b_i \sim \mathrm{Bernoulli}\big(k/n\big). \tag{2}$$

Note that $\widehat{\mathcal{L}}$ accesses on average $k$ loss components $\ell_i$ ($k$ is the expected local sample size), and moreover, $\mathbb{E}\big[\widehat{\mathcal{L}}(\mathbf{w})\big] = \mathcal{L}(\mathbf{w})$ for any $\mathbf{w}$. The goal is to compute local estimates of the Newton's step $\mathbf{p}$ in a communication-efficient manner (i.e., by only sending $O(d)$ parameters from/to a single machine), then combine them into a better global estimate. The gradient has size $O(d)$ so it can be computed exactly within this communication budget (e.g., via map-reduce), however the Hessian has to be approximated locally by each machine. Note that other computational models can be considered, such as those where the global gradient is not computed (and local gradients are used instead).

Under the constraints described above, the most natural strategy is to use directly the Hessian of the locally subsampled loss $\widehat{\mathcal{L}}$ (see, e.g., GIANT [WRKXM18]), resulting in the approximate Newton step $\widehat{\mathbf{p}} = \nabla^{-2}\widehat{\mathcal{L}}(\mathbf{w})\,\nabla\mathcal{L}(\mathbf{w})$. Suppose that we independently construct $m$ i.i.d. copies of this estimate: $\widehat{\mathbf{p}}_1, \ldots, \widehat{\mathbf{p}}_m$ (here, $m$ is the number of machines). Then, for sufficiently large $m$, taking a simple average of the estimates will stop converging to $\mathbf{p}$ because of the inversion bias: $\frac{1}{m}\sum_{t=1}^m \widehat{\mathbf{p}}_t \to \mathbb{E}\big[\widehat{\mathbf{p}}\big] \neq \mathbf{p}$. Figure 1 shows this by plotting the estimation error (in Euclidean distance) of the averaged Newton step estimators, when the weights are uniform and determinantal (for more details and plots, see Appendix C).

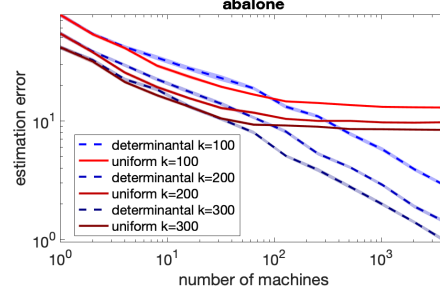

Figure 1: Newton step estimation error versus number of machines, averaged over 100 runs (shading is standard error) for a libsvm dataset [CL11]. More plots in Appendix C.

The only way to reduce the estimation error beyond a certain point is to increase the local sample size $k$ (thereby reducing the inversion bias), which raises the computational cost per machine. Determinantal averaging corrects the inversion bias so that estimation error can always be decreased by adding more machines without increasing the local sample size. From the preceding discussion we can easily show that determinantal averaging leads to an asymptotically consistent estimator. This is a corollary of Theorem 1, as proven in Section 2.

**Corollary 3** *Let $\{\widehat{\mathcal{L}}_t\}_{t=1}^\infty$ be i.i.d. samples of (2) and define $a_t = \det\big(\nabla^2\widehat{\mathcal{L}}_t(\mathbf{w})\big)$. Then:*

$$\frac{\sum_{t=1}^m a_t\,\widehat{\mathbf{p}}_t}{\sum_{t=1}^m a_t} \xrightarrow[m\to\infty]{a.s.} \mathbf{p}, \quad where \quad \widehat{\mathbf{p}}_t = \nabla^{-2}\widehat{\mathcal{L}}_t(\mathbf{w})\,\nabla\mathcal{L}(\mathbf{w}) \ \ and \ \ \mathbf{p} = \nabla^{-2}\mathcal{L}(\mathbf{w})\,\nabla\mathcal{L}(\mathbf{w}).$$

The (unnormalized) determinantal weights can be computed locally in the same time as it takes to compute the Newton estimates so they do not add to the overall cost of the procedure. While this result is only an asymptotic statement, it holds with virtually no assumptions on the loss function (other than twice-differentiability) or the expected local sample size $k$. However, with some additional assumptions we will now establish a convergence guarantee with a finite number of machines $m$ by bounding the estimation error for the determinantal averaging estimator of the Newton step.

In the next result, we use Mahalanobis distance, denoted $\|\mathbf{v}\|_{\mathbf{M}} = \sqrt{\mathbf{v}^\top\mathbf{M}\mathbf{v}}$, to measure the error of the Newton step estimate (i.e., the deviation from optimum $\mathbf{p}$), with $\mathbf{M}$ chosen as the Hessian of $\mathcal{L}$. This choice is motivated by standard convergence analysis of Newton's method, discussed next. This is a corollary of Theorem 2, as explained in Section 3.

**Corollary 4** *For any $\delta, \eta \in (0,1)$ if expected local sample size satisfies $k \geq C\eta^{-2}\mu d^2 \log^3 \frac{d}{\delta}$ then*

$$\left\|\frac{\sum_{t=1}^m a_t\,\widehat{\mathbf{p}}_t}{\sum_{t=1}^m a_t} - \mathbf{p}\right\|_{\nabla^2\mathcal{L}(\mathbf{w})} \leq \frac{\eta}{\sqrt{m}}\cdot\big\|\mathbf{p}\big\|_{\nabla^2\mathcal{L}(\mathbf{w})} \quad with\ probability\ \geq 1-\delta,$$

*where $\mu = \frac{1}{d}\max_i \ell_i''(\mathbf{w}^\top\mathbf{x}_i)\|\mathbf{x}_i\|^2_{\nabla^{-2}\mathcal{L}(\mathbf{w})}$, and $a_t$, $\widehat{\mathbf{p}}_t$ and $\mathbf{p}$ are defined as in Corollary 3.*

We next establish how this error bound impacts the convergence guarantees offered by Newton's method. Note that under our assumptions $\mathcal{L}$ is strongly convex so there is a unique minimizer $\mathbf{w}^* = \operatorname{argmin}_{\mathbf{w}} \mathcal{L}(\mathbf{w})$. We ask how the distance from optimum, $\|\mathbf{w} - \mathbf{w}^*\|$, changes after we make an update $\widetilde{\mathbf{w}} = \mathbf{w} - \widehat{\mathbf{p}}$. For this, we have to assume that the Hessian matrix is $L$-Lipschitz as a function of $\mathbf{w}$. After this standard assumption, a classical analysis of the Newton's method reveals that Corollary 4 implies the following Corollary 6 (proof in Appendix B).

**Assumption 5** *The Hessian is $L$-Lipschitz: $\|\nabla^2\mathcal{L}(\mathbf{w})-\nabla^2\mathcal{L}(\widetilde{\mathbf{w}})\| \leq L\,\|\mathbf{w}-\widetilde{\mathbf{w}}\|$ for any $\mathbf{w},\widetilde{\mathbf{w}} \in \mathbb{R}^d$.*

**Corollary 6** *For any $\delta,\eta \in (0,1)$ if expected local sample size satisfies $k \geq C\eta^{-2}\mu d^2 \log^3\frac{d}{\delta}$ then under Assumption 5 it holds with probability at least $1-\delta$ that*

$$\|\widetilde{\mathbf{w}} - \mathbf{w}^*\| \leq \max\left\{\frac{\eta}{\sqrt{m}}\sqrt{\kappa}\,\|\mathbf{w}-\mathbf{w}^*\|,\ \frac{2L}{\lambda_{\min}}\,\|\mathbf{w}-\mathbf{w}^*\|^2\right\} \quad \textit{for}\ \ \widetilde{\mathbf{w}} = \mathbf{w} - \frac{\sum_{t=1}^{m}a_t\,\widehat{\mathbf{p}}_t}{\sum_{t=1}^{m}a_t},$$

*where $C$, $\mu$, $a_t$ and $\widehat{\mathbf{p}}_t$ are defined as in Corollaries 3 and 4, while $\kappa$ and $\lambda_{\min}$ are the condition number and smallest eigenvalue of $\nabla^2\mathcal{L}(\mathbf{w})$, respectively.*

The bound is a maximum of a linear and a quadratic convergence term. As $m$ goes to infinity and/or $\eta$ goes to 0 the approximation coefficient $\alpha = \frac{\eta}{\sqrt{m}}$ in the linear term disappears and we obtain exact Newton's method, which exhibits quadratic convergence (at least locally around $\mathbf{w}^*$). However, decreasing $\eta$ means increasing $k$ and with it the average computational cost per machine. Thus, to preserve the quadratic convergence while maintaining a computational budget per machine, as the optimization progresses we have to increase the number of machines $m$ while keeping $k$ fixed. This is only possible when we correct for the inversion bias, which is done by determinantal averaging.

## 1.2 Distributed data uncertainty quantification

Here, we consider another example of when computing a compressed linear representation of the inverse matrix is important. Let $\mathbf{X}$ be an $n \times d$ matrix where the rows $\mathbf{x}_i^\top$ represent samples drawn from a population for statistical analysis. The sample covariance matrix $\mathbf{\Sigma} = \frac{1}{n}\mathbf{X}^\top\mathbf{X}$ holds the information about the relations between the features. Assuming that $\mathbf{\Sigma}$ is invertible, the matrix $\mathbf{\Sigma}^{-1}$, also known as the precision matrix, is often used to establish a degree of confidence we have in the data collection [KBCG13]. The diagonal elements of $\mathbf{\Sigma}^{-1}$ are particularly useful since they hold the variance information of each individual feature. Thus, efficiently estimating either the entire diagonal, its trace, or some subset of its entries, is of practical interest [Ste97, WLK$^+$16, BCF09]. We consider the distributed setting where data is separately stored in batches and each local covariance is modeled as:

$$\widehat{\mathbf{\Sigma}} = \frac{1}{k}\sum_{i=1}^{n}b_i\mathbf{x}_i\mathbf{x}_i^\top,\quad\text{where}\quad b_i \sim \text{Bernoulli}(k/n).$$

For each of the local covariances $\widehat{\mathbf{\Sigma}}_1,\dots,\widehat{\mathbf{\Sigma}}_m$, we compute its compressed uncertainty information: $F\big((\widehat{\mathbf{\Sigma}}_t + \frac{\eta}{\sqrt{m}}\mathbf{I})^{-1}\big)$, where we added a small amount of ridge to ensure invertibility[2]. Here, $F(\cdot)$ may for example denote the trace or the vector of diagonal entries. We arrive at the following asymptotically consistent estimator for $F(\mathbf{\Sigma}^{-1})$:

$$\hat{F}_m = \frac{\sum_{t=1}^{m}a_{t,m}F\big((\widehat{\mathbf{\Sigma}}_t + \frac{\eta}{\sqrt{m}}\mathbf{I})^{-1}\big)}{\sum_{t=1}^{m}a_{t,m}},\quad\text{where}\quad a_{t,m} = \det\big(\widehat{\mathbf{\Sigma}}_t + \tfrac{\eta}{\sqrt{m}}\mathbf{I}\big).$$

Note that the ridge term $\frac{\eta}{\sqrt{m}}\mathbf{I}$ decays to zero as $m$ goes to infinity, which is why $\hat{F}_m \to F(\mathbf{\Sigma}^{-1})$. Even though this limit holds for any local sample size $k$, in practice we should choose $k$ sufficiently large so that $\widehat{\mathbf{\Sigma}}$ is well-conditioned. In particular, Theorem 2 implies that if $k \geq 2C\eta^{-2}\mu d^2\log^3\frac{d}{\delta}$, where $\mu = \frac{1}{d}\max_i\|\mathbf{x}_i\|_{\mathbf{\Sigma}^{-1}}^2$, then for $F(\cdot) = \text{tr}(\cdot)$ we have $|\hat{F}_m - \text{tr}(\mathbf{\Sigma}^{-1})| \leq \frac{\eta}{\sqrt{m}}\cdot\text{tr}(\mathbf{\Sigma}^{-1})$ w.p. $1-\delta$.

## 1.3 Related work

Many works have considered averaging strategies for combining distributed estimates, particularly in the context of statistical learning and optimization. This research is particularly important in *federated learning* [KBRR16, KBY$^+$16], where data are spread out across a large network of devices with small local storage and severely constrained communication bandwidth. Using averaging to combine local estimates has been studied in a number of learning settings [MMS$^+$09, MHM10] as

well as for first-order stochastic optimization [ZWLS10, AD11]. For example, [ZDW13] examine the effectiveness of simple uniform averaging of empirical risk minimizers and also propose a bootstrapping technique to reduce the bias.

More recently, distributed averaging methods gained considerable attention in the context of second-order optimization, where the Hessian inversion bias is of direct concern. [SSZ14] propose a distributed approximate Newton-type method (DANE) which under certain assumptions exhibits low bias. This was later extended and improved upon by [ZL15, RKR$^+$16]. The GIANT method of [WRKXM18] most closely follows our setup from Section 1.1, providing non-trivial guarantees for uniform averaging of the Newton step estimates $\widehat{\mathbf{p}}_t$ (except they use with-replacement uniform sampling, whereas we use without-replacement, but that is typically a negligible difference). A related analysis of this approach is provided in the context of ridge regression by [WGM17]. Finally, [ABH17, MLR17, BJKJ17] propose different estimates of the Newton step which exhibit low bias under certain additional assumptions.

Our approach is related to recent developments in determinantal subsampling techniques (e.g., volume sampling), which have been shown to correct the inversion bias in the context of least squares regression [DW17, DWH19]. However, despite recent progress [DW18, DWH18], volume sampling is still far too computationally expensive to be feasible for distributed optimization. Indeed, often uniform sampling is the only practical choice in this context.

With the exception of the expensive volume sampling-based methods, all of the approaches discussed above, even under favorable conditions, use *biased* estimates of the desired solution (e.g., the exact Newton step). Thus, when the number of machines grows sufficiently large, with fixed local sample size, the averaging no longer provides any improvement. This is in contrast to our determinantal averaging, which converges *exactly* to the desired solution and requires no expensive subsampling. Therefore, it can scale with an arbitrarily large number of machines.

## 2    Expectation identities for determinants and adjugates

In this section, we prove Theorem 1 and Corollary 3, establishing that determinantal averaging is asymptotically consistent. To achieve this, we establish a lemma involving two expectation identities.

For a square $n \times n$ matrix $\mathbf{A}$, we use $\mathrm{adj}(\mathbf{A})$ to denote its adjugate, defined as an $n \times n$ matrix whose $(i,j)$th entry is $(-1)^{i+j} \det(\mathbf{A}_{-j,-i})$, where $\mathbf{A}_{-j,-i}$ denotes $\mathbf{A}$ without $j$th row and $i$th column. The adjugate matrix provides a key connection between the inverse and the determinant because for any invertible matrix $\mathbf{A}$, we have $\mathrm{adj}(\mathbf{A}) = \det(\mathbf{A})\mathbf{A}^{-1}$. In the following lemma, we will also use a formula called Sylvester's theorem, relating the adjugate and the determinant: $\det(\mathbf{A} + \mathbf{u}\mathbf{v}^\top) = \det(\mathbf{A}) + \mathbf{v}^\top \mathrm{adj}(\mathbf{A})\mathbf{u}$.

**Lemma 7** *For $\mathbf{A} = \sum_i s_i \mathbf{Z}_i$, where $s_i$ are independently random and $\mathbf{Z}_i$ are square and rank-1,*

$$\text{(a)} \quad \mathbb{E}\big[\det(\mathbf{A})\big] = \det\big(\mathbb{E}[\mathbf{A}]\big) \quad \text{and} \quad \text{(b)} \quad \mathbb{E}\big[\mathrm{adj}(\mathbf{A})\big] = \mathrm{adj}\big(\mathbb{E}[\mathbf{A}]\big).$$

**Proof**  We use induction over the number of components in the sum. If there is only one component, i.e., $\mathbf{A} = s\mathbf{Z}$, then $\det(\mathbf{A}) = 0$ a.s. unless $\mathbf{Z}$ is $1 \times 1$, in which case (a) is trivial, and $(b)$ follows similarly. Now, suppose we showed the hypothesis when the number of components is $n$ and let $\mathbf{A} = \sum_{i=1}^{n+1} s_i \mathbf{Z}_i$. Setting $\mathbf{Z}_{n+1} = \mathbf{u}\mathbf{v}^\top$, we have:

$$\mathbb{E}\big[\det(\mathbf{A})\big] = \mathbb{E}\bigg[ \det\Big( \sum_{i=1}^n s_i \mathbf{Z}_i + s_{n+1}\mathbf{u}\mathbf{v}^\top \Big) \bigg]$$

$$\text{(Sylvester's Theorem)} \quad = \mathbb{E}\bigg[ \det\Big( \sum_{i=1}^n s_i \mathbf{Z}_i \Big) + s_{n+1}\mathbf{v}^\top \mathrm{adj}\Big( \sum_{i=1}^n s_i \mathbf{Z}_i \Big)\mathbf{u} \bigg]$$

$$\text{(inductive hypothesis)} \quad = \det\Big( \mathbb{E}\Big[ \sum_{i=1}^n s_i \mathbf{Z}_i \Big] \Big) + \mathbb{E}[s_{n+1}]\,\mathbf{v}^\top \mathrm{adj}\Big( \mathbb{E}\Big[ \sum_{i=1}^n s_i \mathbf{Z}_i \Big] \Big)\mathbf{u}$$

$$\text{(Sylvester's Theorem)} \quad = \det\Big( \mathbb{E}\Big[ \sum_{i=1}^n s_i \mathbf{Z}_i \Big] + \mathbb{E}[s_{n+1}]\,\mathbf{u}\mathbf{v}^\top \Big) = \det\big(\mathbb{E}[\mathbf{A}]\big),$$

showing (a). Finally, (b) follows by applying (a) to each entry $\mathrm{adj}(\mathbf{A})_{ij} = (-1)^{i+j}\det(\mathbf{A}_{-j,-i})$. ∎

Similar expectation identities for the determinant have been given before [vdV65, DWH19, Der19]. None of them, however, apply to the random matrix $\mathbf{A}$ as defined in Lemma 7, or even to the special case we use for analyzing distributed Newton's method. Also, our proof method is quite different, and somewhat simpler, than those used in prior work. To our knowledge, the extension of determinantal expectation to the adjugate matrix has not previously been pointed out.

We next prove Theorem 1 and Corollary 3 as consequences of Lemma 7.

**Proof of Theorem 1** When $\mathbf{A}$ is invertible, its adjugate is given by $\mathrm{adj}(\mathbf{A}) = \det(\mathbf{A})\mathbf{A}^{-1}$, so the lemma implies that

$$\mathbb{E}\big[\det(\mathbf{A})\big]\big(\mathbb{E}[\mathbf{A}]\big)^{-1} = \det\big(\mathbb{E}[\mathbf{A}]\big)\big(\mathbb{E}[\mathbf{A}]\big)^{-1} = \mathrm{adj}(\mathbb{E}[\mathbf{A}]) = \mathbb{E}\big[\mathrm{adj}(\mathbf{A})\big] = \mathbb{E}\big[\det(\mathbf{A})\mathbf{A}^{-1}\big],$$

from which Theorem 1 follows immediately. ∎

**Proof of Corollary 3** The subsampled Hessian matrix used in Corollary 3 can be written as:

$$\nabla^2\widehat{\mathcal{L}}(\mathbf{w}) = \frac{1}{k}\sum_i b_i \ell_i''(\mathbf{w}^\top\mathbf{x}_i)\,\mathbf{x}_i\mathbf{x}_i^\top \;+\; \lambda\sum_{i=1}^d \mathbf{e}_i\mathbf{e}_i^\top \stackrel{def}{=} \widehat{\mathbf{H}},$$

so, letting $\widehat{\mathbf{H}}_t = \nabla^2\widehat{\mathcal{L}}_t(\mathbf{w})$, Corollary 3 follows from Theorem 1 and the law of large numbers:

$$\frac{\sum_{t=1}^m a_t\,\widehat{\mathbf{p}}_t}{\sum_{t=1}^m a_t} = \frac{\frac{1}{m}\sum_{t=1}^m \det\big(\widehat{\mathbf{H}}_t\big)\widehat{\mathbf{H}}_t^{-1}\nabla\mathcal{L}(\mathbf{w})}{\frac{1}{m}\sum_{t=1}^m \det\big(\widehat{\mathbf{H}}_t\big)} \xrightarrow[m\to\infty]{} \frac{\mathbb{E}\big[\det(\widehat{\mathbf{H}})\widehat{\mathbf{H}}^{-1}\big]}{\mathbb{E}\big[\det(\widehat{\mathbf{H}})\big]}\nabla\mathcal{L}(\mathbf{w}) = \nabla^{-2}\mathcal{L}(\mathbf{w})\,\nabla\mathcal{L}(\mathbf{w}),$$

which concludes the proof. ∎

## 3 Finite-sample convergence analysis

In this section, we prove Theorem 2 and Corollary 4, establishing that determinantal averaging exhibits a $1/\sqrt{m}$ convergence rate, where $m$ is the number of sampled matrices (or the number of machines in distributed Newton's method). For this, we need a tool from random matrix theory.

**Lemma 8 (Matrix Bernstein [Tro12])** *Consider a finite sequence $\{\mathbf{X}_i\}$ of independent, random, self-adjoint matrices with dimension $d$ such that $\mathbb{E}[\mathbf{X}_i] = \mathbf{0}$ and $\lambda_{\max}(\mathbf{X}_i) \le R$ almost surely. If the sequence satisfies $\big\|\sum_i \mathbb{E}[\mathbf{X}_i^2]\big\| \le \sigma^2$, then the following inequality holds for all $x \ge 0$:*

$$\Pr\Big(\lambda_{\max}\Big(\sum_i \mathbf{X}_i\Big) \ge x\Big) \le \begin{cases} d\,e^{-\frac{x^2}{4\sigma^2}} & \text{for } x \le \frac{\sigma^2}{R}; \\ d\,e^{-\frac{x}{4R}} & \text{for } x \ge \frac{\sigma^2}{R}. \end{cases}$$

The key component of our analysis is bounding the moments of the determinant and adjugate of a certain class of random matrices. This has to be done carefully, because higher moments of the determinant grow more rapidly than, e.g., for a sub-gaussian random variable. For this result, we do not require that the individual components $\mathbf{Z}_i$ of matrix $\mathbf{A}$ be rank-1, but we impose several additional boundedness assumptions. In the proof below we apply the concentration inequality of Lemma 8 twice: first to the random matrix $\mathbf{A}$ itself, and then also to its trace, which allows finer control over the determinant.

**Lemma 9** *Let $\mathbf{A} = \frac{1}{\gamma}\sum_i b_i\mathbf{Z}_i + \mathbf{B}$, where $b_i \sim \mathrm{Bernoulli}(\gamma)$ are independent, whereas $\mathbf{Z}_i$ and $\mathbf{B}$ are $d \times d$ psd matrices such that $\|\mathbf{Z}_i\| \le \epsilon$ for all $i$ and $\mathbb{E}[\mathbf{A}] = \mathbf{I}$. If $\gamma \ge 8\epsilon d\eta^{-2}(p + \ln d)$ for $0 < \eta \le 0.25$ and $p \ge 2$, then*

$$\text{(a)} \quad \mathbb{E}\Big[\big|\det(\mathbf{A}) - 1\big|^p\Big]^{\frac{1}{p}} \le 5\eta \quad \text{and} \quad \text{(b)} \quad \mathbb{E}\Big[\big\|\mathrm{adj}(\mathbf{A}) - \mathbf{I}\big\|^p\Big]^{\frac{1}{p}} \le 9\eta.$$

**Proof** We start by proving (a). Let $X = \det(\mathbf{A}) - 1$ and denote $\mathbf{1}_{[a,b]}$ as the indicator variable of the event that $X \in [a,b]$. Since $\det(\mathbf{A}) \ge 0$, we have:

$$\mathbb{E}\big[|X|^p\big] = \mathbb{E}\big[(-X)^p \cdot \mathbf{1}_{[-1,0]}\big] + \mathbb{E}\big[X^p \cdot \mathbf{1}_{[0,\infty]}\big]$$

$$\le \eta^p + \int_\eta^1 px^{p-1}\Pr(-X \ge x)dx + \int_0^\infty px^{p-1}\Pr(X \ge x)dx. \tag{3}$$

Thus it suffices to bound the two integrals. We will start with the first one. Let $\mathbf{X}_i = (1 - \frac{b_i}{\gamma})\mathbf{Z}_i$. We use the matrix Bernstein inequality to control the extreme eigenvalues of the matrix $\mathbf{I} - \mathbf{A} = \sum_i \mathbf{X}_i$ (note that matrix $\mathbf{B}$ cancels out because $\mathbf{I} = \mathbb{E}[\mathbf{A}] = \sum_i \mathbf{Z}_i + \mathbf{B}$). To do this, observe that $\|\mathbf{X}_i\| \leq \epsilon/\gamma$ and, moreover, $\mathbb{E}[(1 - \frac{b_i}{\gamma})^2] = \frac{1}{\gamma} - 1 \leq \frac{1}{\gamma}$, so:

$$\Big\| \sum_i \mathbb{E}[\mathbf{X}_i^2] \Big\| = \Big\| \sum_i \mathbb{E}[(1 - \tfrac{b_i}{\gamma})^2]\mathbf{Z}_i^2 \Big\| \leq \frac{1}{\gamma} \cdot \Big\| \sum_i \mathbf{Z}_i^2 \Big\| \leq \frac{\epsilon}{\gamma} \cdot \Big\| \sum_i \mathbf{Z}_i \Big\| \leq \frac{\epsilon}{\gamma}.$$

Thus, applying Lemma 8 we conclude that for any $z \in \big[\frac{\eta}{\sqrt{2d}}, 1\big]$:

$$\Pr\Big( \|\mathbf{I} - \mathbf{A}\| \geq z \Big) \leq 2d\, \mathrm{e}^{-\frac{z^2 \gamma}{4\epsilon}} \leq 2\mathrm{e}^{\ln(d) - z^2 \frac{2d}{\eta^2}(p + \ln d)} \leq 2\mathrm{e}^{-z^2 \frac{2dp}{\eta^2}}. \tag{4}$$

Conditioning on the high-probability event given by (4) leads to the lower bound $\det(\mathbf{A}) \geq (1 - z)^d$ which is very loose. To improve on it, we use the following inequality, where $\delta_1, \ldots, \delta_d$ denote the eigenvalues of $\mathbf{I} - \mathbf{A}$:

$$\det(\mathbf{A})\mathrm{e}^{\mathrm{tr}(\mathbf{I}-\mathbf{A})} = \prod_i (1 - \delta_i)\mathrm{e}^{\delta_i} \geq \prod_i (1 - \delta_i)(1 + \delta_i) = \prod_i (1 - \delta_i^2).$$

Thus we obtain a tighter bound when $\det(\mathbf{A})$ is multiplied by $\mathrm{e}^{\mathrm{tr}(\mathbf{I}-\mathbf{A})}$, and now it suffices to upper bound the latter. This is a simple application of the scalar Bernstein's inequality (Lemma 8 with $d = 1$) for the random variables $X_i = \mathrm{tr}(\mathbf{X}_i) \leq \epsilon/\gamma \leq \frac{\eta^2}{8dp}$, which satisfy $\sum_i \mathbb{E}[X_i^2] \leq \frac{\epsilon}{\gamma}\mathrm{tr}(\sum_i \mathbf{Z}_i) \leq \frac{\epsilon d}{\gamma} \leq \frac{\eta^2}{8p}$. Thus the scalar Bernstein's inequality states that

$$\max\Big\{ \Pr\big(\mathrm{tr}(\mathbf{A} - \mathbf{I}) \geq y\big),\ \Pr\big(\mathrm{tr}(\mathbf{A} - \mathbf{I}) \leq -y\big) \Big\} \leq \begin{cases} \mathrm{e}^{-y^2 \frac{2p}{\eta^2}} & \text{for } y \leq d; \\ \mathrm{e}^{-y \frac{2dp}{\eta^2}} & \text{for } y \geq d. \end{cases} \tag{5}$$

Setting $y = \frac{x}{2}$ and $z = \sqrt{\frac{x}{2d}}$ and taking a union bound over the appropriate high-probability events given by (4) and (5), we conclude that for any $x \in [\eta, 1]$:

$$\det(\mathbf{A}) \geq (1 - z^2)^d \exp\big(\mathrm{tr}(\mathbf{A} - \mathbf{I})\big) \geq \big(1 - \tfrac{x}{2}\big)\mathrm{e}^{-\frac{x}{2}} \geq 1 - x, \quad \text{with prob. } 1 - 3\mathrm{e}^{-x^2 \frac{p}{2\eta^2}}.$$

Thus, for $X = \det(\mathbf{A}) - 1$ and $x \in [\eta, 1]$ we obtain that $\Pr(-X \geq x) \leq 3\mathrm{e}^{-x^2 \frac{p}{2\eta^2}}$, and consequently,

$$\int_\eta^1 px^{p-1} \Pr\big(-X \geq x\big)dx \leq 3p \int_\eta^1 x^{p-1}\mathrm{e}^{-x^2 \frac{p}{2\eta^2}}dx \leq 3p\sqrt{\pi \frac{2\eta^2}{p}} \cdot \int_{-\infty}^\infty |x|^{p-1} \frac{\mathrm{e}^{-x^2 \frac{p}{2\eta^2}}}{\sqrt{2\pi\eta^2/p}}dx$$

$$\leq 3\sqrt{2\pi\eta^2 p} \cdot \big(\tfrac{\eta^2}{p}p\big)^{\frac{p-1}{2}} = 3\sqrt{2\pi p} \cdot \eta^p.$$

We now move on to bounding the remaining integral from (3). Since determinant is the product of eigenvalues, we have $\det(\mathbf{A}) = \det(\mathbf{I} + \mathbf{A} - \mathbf{I}) \leq \mathrm{e}^{\mathrm{tr}(\mathbf{A}-\mathbf{I})}$, so we can use the Bernstein bound of (5) w.r.t. $\mathbf{A} - \mathbf{I}$. It follows that:

$$\int_0^\infty px^{p-1}\Pr(X \geq x)dx \leq \int_0^\infty px^{p-1} \Pr\big(\mathrm{e}^{\mathrm{tr}(\mathbf{A}-\mathbf{I})} \geq 1 + x\big)dx$$

$$\leq \int_0^{\mathrm{e}^d - 1} px^{p-1}\mathrm{e}^{-\ln^2(1+x)\frac{2p}{\eta^2}}dx + \int_{\mathrm{e}^d - 1}^\infty px^{p-1}\mathrm{e}^{-\ln(1+x)\frac{2dp}{\eta^2}}dx$$

$$\leq \int_0^{\mathrm{e} - 1} px^{p-1}\mathrm{e}^{-\ln^2(1+x)\frac{2p}{\eta^2}}dx + \int_{\mathrm{e} - 1}^\infty px^{p-1}\mathrm{e}^{-\ln(1+x)\frac{2p}{\eta^2}}dx,$$

because $\ln^2(1 + x) \geq \ln(1 + x)$ for $x \geq \mathrm{e} - 1$. Note that $\ln^2(1 + x) \geq x^2/4$ for $x \in [0, \mathrm{e} - 1]$, so

$$\int_0^{\mathrm{e} - 1} px^{p-1}\mathrm{e}^{-\ln^2(1+x)\frac{2p}{\eta^2}}dx \leq \int_0^{\mathrm{e} - 1} px^{p-1}\mathrm{e}^{-x^2\frac{p}{2\eta^2}}dx \leq \sqrt{2\pi p} \cdot \eta^p.$$

In the interval $x \in [\mathrm{e} - 1, \infty]$, we have:

$$\int_{\mathrm{e}-1}^\infty px^{p-1}\mathrm{e}^{-\ln(1+x)\frac{2p}{\eta^2}}dx = p\int_{\mathrm{e}-1}^\infty \mathrm{e}^{(p-1)\ln(x) - \ln(1+x)\frac{2p}{\eta^2}}dx \leq p\int_{\mathrm{e}-1}^\infty \mathrm{e}^{-\ln(1+x)\frac{p}{\eta^2}}dx$$

$$\leq p\int_1^\infty \big(\tfrac{1}{1+x}\big)^{\frac{p}{\eta^2}}dx = \frac{p}{\frac{p}{\eta^2} - 1}\big(\tfrac{1}{2}\big)^{\frac{p}{\eta^2} - 1} \leq p \cdot \Big(\big(\tfrac{1}{2}\big)^{\frac{1}{\eta^2}}\Big)^p \leq p \cdot \big(\eta^2\big)^p,$$

where the last inequality follows because $(\frac{1}{2})^{\frac{1}{\eta^2}} \leq \eta^2$. Noting that $(1 + 4\sqrt{2\pi p} + p)^{\frac{1}{p}} \leq 5$ for any $p \geq 2$ concludes the proof of (a). The proof of (b), given in Appendix A, follows similarly as above because for any positive definite $\mathbf{A}$ we have $\frac{\det(\mathbf{A})}{\lambda_{\max}(\mathbf{A})} \cdot \mathbf{I} \preceq \mathrm{adj}(\mathbf{A}) \preceq \frac{\det(\mathbf{A})}{\lambda_{\min}} \cdot \mathbf{I}$. ∎

Having obtained bounds on the higher moments, we can now convert them to convergence with high probability for the average of determinants and the adjugates. Since determinant is a scalar variable, this follows by using standard arguments. On the other hand, for the adjugate matrix we require a somewhat less standard matrix extension of the Khintchine/Rosenthal inequalities (see Appendix A).

**Corollary 10** *There is $C > 0$ s.t. for $\mathbf{A}$ as in Lemma 9 with all $\mathbf{Z}_i$ rank-1 and $\gamma \geq C\epsilon d\eta^{-2} \log^3 \frac{d}{\delta}$,*

$$(a) \ \Pr\left(\left|\frac{1}{m}\sum_{t=1}^{m}\det(\mathbf{A}_t) - 1\right| \geq \frac{\eta}{\sqrt{m}}\right) \leq \delta \quad and \quad (b) \ \Pr\left(\left\|\frac{1}{m}\sum_{t=1}^{m}\mathrm{adj}(\mathbf{A}_t) - \mathbf{I}\right\| \geq \frac{\eta}{\sqrt{m}}\right) \leq \delta,$$

*where $\mathbf{A}_1, \ldots, \mathbf{A}_m$ are independent copies of $\mathbf{A}$.*

We are ready to show the convergence rate of determinantal averaging, which follows essentially by upper/lower bounding the enumerator and denominator separately, using Corollary 10.

**Proof of Theorem 2** We will apply Corollary 10 to the matrices $\mathbf{A}_t = \mathbf{H}^{-\frac{1}{2}}\widehat{\mathbf{H}}_t\mathbf{H}^{-\frac{1}{2}}$. Note that $\mathbf{A}_t = \frac{n}{k}\sum_i b_i\widetilde{\mathbf{Z}}_i + \lambda\mathbf{H}^{-1}$, where each $\widetilde{\mathbf{Z}}_i = \frac{1}{n}\mathbf{H}^{-\frac{1}{2}}\mathbf{Z}_i\mathbf{H}^{-\frac{1}{2}}$ satisfies $\|\widetilde{\mathbf{Z}}_i\| \leq \mu \cdot d/n$. Therefore, Corollary 10 guarantees that for $\frac{k}{n} \geq C\frac{\mu d}{n}d\eta^{-2}\log^3\frac{d}{\delta}$, with probability $1 - \delta$ the following average of determinants is concentrated around 1:

$$Z \overset{def}{=} \frac{1}{m}\sum_t \frac{\det(\widehat{\mathbf{H}}_t)}{\det(\mathbf{H})} = \frac{1}{m}\sum_t \det\left(\mathbf{H}^{-\frac{1}{2}}\widehat{\mathbf{H}}_t\mathbf{H}^{-\frac{1}{2}}\right) \in [1 - \alpha, 1 + \alpha] \quad \text{for } \alpha = \frac{\eta}{\sqrt{m}},$$

along with a corresponding bound for the adjugate matrices. We obtain that with probability $1 - 2\delta$,

$$\left\|\frac{\sum_{t=1}^{m}\mathrm{adj}(\mathbf{A}_t)}{\sum_{t=1}^{m}\det(\mathbf{A}_t)} - \mathbf{I}\right\| \leq \left\|\frac{1}{m}\sum_t\mathrm{adj}(\mathbf{A}_t) - Z\mathbf{I}\right\|/Z$$

$$\text{(Corollary 10a)} \quad \leq \frac{1}{1-\alpha}\left\|\frac{1}{m}\sum_t\mathrm{adj}(\mathbf{A}_t) - \mathbf{I}\right\| + \frac{\alpha}{1-\alpha}$$

$$\text{(Corollary 10b)} \quad \leq \frac{\alpha}{1-\alpha} + \frac{\alpha}{1-\alpha}.$$

It remains to multiply the above expressions by $\mathbf{H}^{-\frac{1}{2}}$ from both sides to recover the desired estimator:

$$\frac{\sum_{t=1}^{m}\det(\widehat{\mathbf{H}}_t)\widehat{\mathbf{H}}_t^{-1}}{\sum_{t=1}^{m}\det(\widehat{\mathbf{H}}_t)} = \mathbf{H}^{-\frac{1}{2}}\frac{\sum_{t=1}^{m}\mathrm{adj}(\mathbf{A}_t)}{\sum_{t=1}^{m}\det(\mathbf{A}_t)}\mathbf{H}^{-\frac{1}{2}} \preceq \mathbf{H}^{-\frac{1}{2}}\left(1 + \frac{2\alpha}{1-\alpha}\right)\mathbf{I}\,\mathbf{H}^{-\frac{1}{2}} = \left(1 + \frac{2\alpha}{1-\alpha}\right)\mathbf{H}^{-1},$$

and the lower bound follows identically. Appropriately adjusting the constants concludes the proof. ∎

As an application of the above result, we show how this allows us to bound the estimation error in distributed Newton's method, when using determinantal averaging.

**Proof of Corollary 4** Follows from Theorem 2 by setting $\mathbf{Z}_i = \ell_i''(\mathbf{w}^\top\mathbf{x}_i)\mathbf{x}_i\mathbf{x}_i^\top$ and $\mathbf{B} = \lambda\mathbf{I}$. Note that the assumptions imply that $\|\mathbf{Z}_i\| \leq \mu$, so invoking the theorem and denoting $\mathbf{g}$ as $\nabla\mathcal{L}(\mathbf{w})$, with probability $1 - \delta$ we have

$$\left\|\frac{\sum_{t=1}^{m}a_t\widehat{\mathbf{p}}_t}{\sum_{t=1}^{m}a_t} - \mathbf{p}\right\|_{\mathbf{H}} = \left\|\mathbf{H}^{\frac{1}{2}}\left(\frac{\sum_{t=1}^{m}\det(\widehat{\mathbf{H}}_t)\widehat{\mathbf{H}}_t^{-1}}{\sum_{t=1}^{m}\det(\widehat{\mathbf{H}}_t)} - \mathbf{H}^{-1}\right)\mathbf{H}^{\frac{1}{2}}\,\mathbf{H}^{-\frac{1}{2}}\mathbf{g}\right\|$$

$$\leq \left\|\mathbf{H}^{\frac{1}{2}}\left(\frac{\sum_{t=1}^{m}\det(\widehat{\mathbf{H}}_t)\widehat{\mathbf{H}}_t^{-1}}{\sum_{t=1}^{m}\det(\widehat{\mathbf{H}}_t)} - \mathbf{H}^{-1}\right)\mathbf{H}^{\frac{1}{2}}\right\| \cdot \left\|\mathbf{H}^{-\frac{1}{2}}\mathbf{g}\right\|$$

$$\text{(Theorem 2)} \quad \leq \left\|\mathbf{H}^{\frac{1}{2}}\frac{\eta}{\sqrt{m}}\mathbf{H}^{-1}\mathbf{H}^{\frac{1}{2}}\right\| \cdot \|\mathbf{p}\|_{\mathbf{H}} = \frac{\eta}{\sqrt{m}} \cdot \|\mathbf{p}\|_{\mathbf{H}},$$

which completes the proof of the corollary. ∎

# 4 Conclusions and future directions

We proposed a novel method for correcting the inversion bias in distributed Newton's method. Our approach, called determinantal averaging, can also be applied more broadly to distributed estimation of other linear functions of the inverse Hessian or an inverse covariance matrix. We show that estimators produced by determinantal averaging are asymptotically consistent, and we provide bounds on the estimation error by developing new moment bounds on the determinant of a random matrix.

Further empirical evaluation of determinantal averaging, both in the context of distributed optimization and other tasks involving inverse estimation, is an important direction for future work. Our preliminary experiments suggest that the bias-correction of determinantal averaging comes at a price of additional variance in the estimators. This leads to a natural open problem: find the optimal balance between bias and variance in weighted averaging for distributed inverse estimation. Finally, note that we construct our Newton estimates using local Hessian and global gradient. In some settings it is more practical to use local approximations for both the Hessian and the gradient. Whether or not determinantal averaging corrects the bias in this case remains open.

### Acknowledgements

MWM would like to acknowledge ARO, DARPA, NSF and ONR for providing partial support of this work. Also, MWM and MD thank the NSF for funding via the NSF TRIPODS program. Part of this work was done while MD and MWM were visiting the Simons Institute for the Theory of Computing.

## Footnotes

[1]Clearly, one would not actually compute the inverse of the Hessian explicitly [XRKM17, YXRKM18]. We describe it this way for simplicity. Our results hold whether or not the inverse operator is computed explicitly.

[2] Since the ridge term vanishes as $m$ goes to infinity, we are still estimating the ridge-free quantity $F(\mathbf{\Sigma}^{-1})$.

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
