[Supplementary Material]

## A  Omitted proofs from Section 3

In Section 3, we stated Lemma 9 and proved the first part of it (a moment bound for the determinant). Here, we provide the proof of the second part (a moment bound for the adjugate).

**Lemma 11 (Lemma 9b restated)** *Let* $\mathbf{A} = \frac{1}{\gamma}\sum_i b_i \mathbf{Z}_i + \mathbf{B}$, *where* $b_i \sim \text{Bernoulli}(\gamma)$ *are independent, whereas* $\mathbf{Z}_i$ *and* $\mathbf{B}$ *are* $d \times d$ *psd matrices such that* $\|\mathbf{Z}_i\| \leq \epsilon$ *for all* $i$ *and* $\mathbb{E}[\mathbf{A}] = \mathbf{I}$. *If* $\gamma \geq 8\epsilon d\eta^{-2}(p + \ln d)$ *for* $0 < \eta \leq 0.25$ *and* $p \geq 2$, *then*

$$\mathbb{E}\Big[\big\|\operatorname{adj}(\mathbf{A}) - \mathbf{I}\big\|^p\Big]^{\frac{1}{p}} \leq 9\eta.$$

**Proof** Let $\lambda_{\max}$ and $\lambda_{\min}$ denote the largest and smallest eigenvalue of $\operatorname{adj}(\mathbf{A})$. We have

$$\mathbb{E}\big[\|\operatorname{adj}(\mathbf{A}) - \mathbf{I}\|^p\big] = \int_0^\infty px^{p-1}\Pr\big(\|\operatorname{adj}(\mathbf{A}) - \mathbf{I}\| \geq x\big)dx$$

$$\leq \eta^p + \int_\eta^\infty px^{p-1}\Big(\Pr(\lambda_{\max} \geq 1 + x) + \Pr(\lambda_{\min} \leq 1 - x)\Big)dx.$$

We will now bound the two probabilities. Let $\delta_{\max}$ and $\delta_{\min}$ denote the largest and smallest eigenvalue of matrix $\mathbf{A} - \mathbf{I}$. Recall the following concentration bounds implied by Lemma 8 (see the first part of the proof of Lemma 9):

$$\max\Big\{\Pr\big(\operatorname{tr}(\mathbf{A} - \mathbf{I}) \geq y\big),\ \Pr\big(\operatorname{tr}(\mathbf{A} - \mathbf{I}) \leq -y\big)\Big\} \leq \begin{cases} e^{-y^2\frac{2p}{\eta^2}} & \text{for } y \in [0, d]; \\ e^{-y\frac{2dp}{\eta^2}} & \text{for } y \geq d, \end{cases} \tag{6}$$

$$\max\Big\{\Pr\big(\delta_{\max} \geq z\big),\ \Pr\big(\delta_{\min} \leq -z\big)\Big\} \leq \begin{cases} e^{-z^2\frac{2p}{\eta^2}} & \text{for } z \in [0, \frac{\eta}{\sqrt{2d}}]; \\ e^{-z^2\frac{2dp}{\eta^2}} & \text{for } z \in [\frac{\eta}{\sqrt{2d}}, 1]; \\ e^{-z\frac{2dp}{\eta^2}} & \text{for } z \geq 1. \end{cases} \tag{7}$$

From the formula $\operatorname{adj}(\mathbf{A}) = \det(\mathbf{A})\mathbf{A}^{-1}$ it follows that $\lambda_{\max} \leq \frac{\det(\mathbf{A})}{1 + \delta_{\min}} \leq \frac{e^{\operatorname{tr}(\mathbf{A}-\mathbf{I})}}{1 + \delta_{\min}}$ so we have

$$\Pr\big(\lambda_{\max} \geq 1 + x\big) \leq \Pr\left(\frac{e^{\operatorname{tr}(\mathbf{A}-\mathbf{I})}}{1 + \delta_{\min}} \geq 1 + x\right)$$

$$= \Pr\left(\operatorname{tr}(\mathbf{A} - \mathbf{I}) + \ln\frac{1}{1 + \delta_{\min}} \geq \ln(1 + x)\right)$$

$$\leq \Pr\left(\operatorname{tr}(\mathbf{A} - \mathbf{I}) \geq \frac{2}{3}\cdot\ln(1 + x)\right) + \Pr\left(\ln\frac{1}{1 + \delta_{\min}} \geq \frac{1}{3}\cdot\ln(1 + x)\right)$$

$$= \Pr\left(\operatorname{tr}(\mathbf{A} - \mathbf{I}) \geq \frac{2}{3}\cdot\ln(1 + x)\right) + \Pr\left(\delta_{\min} \leq \frac{1}{(1 + x)^{\frac{1}{3}}} - 1\right).$$

$$\leq \begin{cases} e^{-\ln^2(1+x)\frac{8p}{9\eta^2}} + e^{-(1-(\frac{1}{1+x})^{\frac{1}{3}})^2\frac{2p}{\eta^2}} \leq 2e^{-x^2\frac{p}{20\eta^2}} & \text{for } x \in [0, e-1], \\ e^{-\ln(1+x)\frac{4p}{3\eta^2}} + e^{-\frac{1}{16}\frac{2dp}{\eta^2}} \leq 2e^{-\ln(1+x)\frac{p}{8\eta^2}} & \text{for } x \in [e-1, e^d-1]. \end{cases}$$

For $x \geq e^d - 1$, since $\lambda_{\max} \leq (1 + \delta_{\max})^d \leq e^{d\delta_{\max}}$ and $\ln(1 + x) \geq d$, we have:

$$\Pr\big(\lambda_{\max} \geq 1 + x\big) \leq \Pr\big(e^{d\delta_{\max}} \geq 1 + x\big) = \Pr\big(\delta_{\max} \geq \ln(1 + x)/d\big) \leq e^{-\ln(1+x)\frac{2p}{\eta^2}}.$$

Next, we use the fact that for $\delta = \max\big\{|\delta_{\max}|, |\delta_{\min}|\big\}$ we have:

$$\lambda_{\min} \geq \frac{\det(\mathbf{A})}{1 + \delta_{\max}} \geq \frac{(1 - \delta^2)^d}{(1 + \delta_{\max})e^{\operatorname{tr}(\mathbf{I}-\mathbf{A})}} \geq (1 - \delta)(1 - d\delta^2)\big(1 - \operatorname{tr}(\mathbf{I} - \mathbf{A})\big),$$

so for $x \in [\eta, 1]$ we have:

$$\Pr\big(\lambda_{\min} \leq 1 - x\big) \leq \Pr(\delta \geq x/3) + \Pr(\delta^2 \geq x/3d) + \Pr\big(\operatorname{tr}(\mathbf{I} - \mathbf{A}) \geq x/3\big)$$

$$\leq 2e^{-x^2\frac{2dp}{9\eta^2}} + 2e^{-\frac{x}{3d}\frac{2dp}{\eta^2}} + e^{-x^2\frac{2p}{\eta^2}} \leq 5e^{-x^2\frac{2p}{9\eta^2}}.$$

Putting everything together we obtain that:

$$\mathbb{E}\big[\|\operatorname{adj}(\mathbf{A})-\mathbf{I}\|^p\big] \le \eta^p + \int_{\eta}^{e-1} px^{p-1}\, 7\mathrm{e}^{-x^2\frac{p}{20\eta^2}}\,\mathrm{d}x + \int_{e-1}^{\infty} px^{p-1} 3\mathrm{e}^{-\ln(1+x)\frac{p}{8\eta^2}}\,\mathrm{d}x$$

$$\le \eta^p + 7\sqrt{20\pi p}\,\eta^p + \frac{3p}{\frac{p}{16\eta^2}-1}\Big(\tfrac{1}{2}\Big)^{\frac{p}{16\eta^2}-1}$$

$$\le \eta^p + 7\sqrt{20\pi p}\,\eta^p + 6(3\eta)^p \ \le\ (9\eta)^p,$$

which completes the proof. ∎

As a consequence of the moment bounds shown in Lemma 9, we establish convergence with high probability for the average of determinants and the adjugates. For the adjugate matrix, we require a matrix variant of the Khintchine/Rosenthal inequalities.

**Lemma 12 ([GCT12])** *Suppose that $p \ge 2$ and $r = \max\{p, 2\log d\}$.[2] Consider a finite sequence $\{\mathbf{X}_i\}$ of independent, symmetrically random, self-adjoint matrices with dimension $d \times d$. Then,*

$$\mathbb{E}\Big[\|\sum_i \mathbf{X}_i\|^p\Big]^{\frac{1}{p}} \le \sqrt{er}\,\Big\|\sum_i \mathbb{E}[\mathbf{X}_i^2]\Big\|^{\frac{1}{2}} + 2er\,\mathbb{E}\big[\max_i \|\mathbf{X}_i\|^p\big]^{\frac{1}{p}}.$$

**Corollary 13 (Corollary 10 restated)** *There is $C > 0$ s.t. for $\mathbf{A}$ as in Lemma 9 with all $\mathbf{Z}_i$ rank-$1$ and $\gamma \ge C\epsilon d\eta^{-2}\log^3\frac{d}{\delta}$,*

$$(a)\ \Pr\Big(\Big|\frac{1}{m}\sum_{t=1}^{m}\det(\mathbf{A}_t)-1\Big| \ge \frac{\eta}{\sqrt{m}}\Big) \le \delta \quad and \quad (b)\ \Pr\Big(\Big\|\frac{1}{m}\sum_{t=1}^{m}\operatorname{adj}(\mathbf{A}_t)-\mathbf{I}\Big\| \ge \frac{\eta}{\sqrt{m}}\Big) \le \delta,$$

*where $\mathbf{A}_1,\ldots,\mathbf{A}_m$ are independent copies of $\mathbf{A}$.*

**Proof** Applying Lemma 9 to the matrix $\mathbf{A}$, for appropriate $C$ and any fixed $p \ge 2$, if $\gamma \ge C\epsilon d\sigma^{-2}(p + \ln d)$, then for any $s \in [2, p]$ we have $\mathbb{E}\big[\|\operatorname{adj}(\mathbf{A}_t)-\mathbf{I}\|^s\big] \le \sigma^s$. With the additional assumption that $\mathbf{Z}_i$'s are rank-$1$, Theorem 7 implies that $\mathbb{E}\big[\operatorname{adj}(\mathbf{A}_t)\big] = \mathbf{I}$, so by a standard symmetrization argument, where $r_t$ denote independent Rademacher random variables,

$$\mathbb{E}\Big[\Big\|\frac{1}{m}\sum_{t=1}^{m}\operatorname{adj}(\mathbf{A}_t)-\mathbf{I}\Big\|^p\Big]^{\frac{1}{p}} \le 2\cdot\mathbb{E}\Big[\Big\|\sum_t \frac{r_t}{m}\big(\operatorname{adj}(\mathbf{A}_t)-\mathbf{I}\big)\Big\|^p\Big]^{\frac{1}{p}}.$$

Applying Lemma 12 to the matrices $\mathbf{X}_t = \frac{1}{m}\mathbf{Y}_t$, where $\mathbf{Y}_t = r_t\big(\operatorname{adj}(\mathbf{A}_t)-\mathbf{I}\big)$, we obtain that:

$$\mathbb{E}\Big[\Big\|\frac{1}{m}\sum_t \mathbf{Y}_t\Big\|^p\Big]^{\frac{1}{p}} \le \sqrt{er}\,\Big\|m\cdot\frac{1}{m^2}\mathbb{E}[\mathbf{Y}_t^2]\Big\|^{\frac{1}{2}} + \frac{2er}{m}\,\mathbb{E}\Big[\sum_{i=1}^{m}\|\mathbf{Y}_i\|^p\Big]^{\frac{1}{p}}$$

$$\le \sqrt{\frac{er}{m}}\cdot\mathbb{E}\big[\|\mathbf{Y}\|^2\big]^{\frac{1}{2}} + \frac{2er}{m}\Big(m\cdot\mathbb{E}\big[\|\mathbf{Y}\|^p\big]\Big)^{\frac{1}{p}}$$

$$\le \Big(\sqrt{\frac{er}{m}} + \frac{2er}{m^{1-\frac{1}{p}}}\Big)\cdot\sigma \le C'\cdot\frac{p\sigma}{\sqrt{m}},$$

for $p \ge 2\log d$ and $C'$ chosen appropriately. Now Markov's inequality yields:

$$\Pr\Big(\Big\|\frac{1}{m}\sum_t \operatorname{adj}(\mathbf{A}_t)-\mathbf{I}\Big\| \ge \alpha\Big) \le \alpha^{-p}\cdot\mathbb{E}\Big[\Big\|\frac{1}{m}\sum_t \operatorname{adj}(\mathbf{A}_t)-\mathbf{I}\Big\|^p\Big] \le \Big(\frac{2C'p\sigma}{\alpha\sqrt{m}}\Big)^p.$$

Setting $\alpha = \frac{\eta}{\sqrt{m}}$, $\sigma = \frac{\eta}{4C'p}$ and $p = 2\lceil\max\{\log d, \log\frac{1}{\delta}\}\rceil$, the above bound becomes $(\tfrac{1}{2})^p \le \delta$ for $k \ge C''\mu d^2\eta^{-2}(\log^3\frac{1}{\delta} + \log^3 d)$. Showing the analogous result for the average of determinants of matrices $\mathbf{A}_t$ instead of the adjugates follows identically, except that Lemma 12 can be replaced with the standard scalar Rosenthal's inequality. ∎

## B Proof of Newton convergence

Here, we provide a proof of Corollary 6, which describes the convergence guarantees for the approximate Newton step obtained via determinantal averaging. It suffices to show the following lemma.

**Lemma 14** *Let loss $\mathcal{L}$ be defined as in* (1) *and assume its Hessian is L-Lipschitz (Assumption 5). If*

$$\left\|\widehat{\mathbf{p}} - \mathbf{p}^*\right\|_{\nabla^2\mathcal{L}(\mathbf{w})} \leq \alpha \left\|\mathbf{p}^*\right\|_{\nabla^2\mathcal{L}(\mathbf{w})}, \quad \text{where} \quad \mathbf{p}^* = \nabla^{-2}\mathcal{L}(\mathbf{w})\,\nabla\mathcal{L}(\mathbf{w}),$$

*then the approximate Newton step $\widetilde{\mathbf{w}} = \mathbf{w} - \widehat{\mathbf{p}}$ satisfies:*

$$\|\widetilde{\mathbf{w}} - \mathbf{w}^*\| \leq \max\left\{\alpha\sqrt{\kappa}\,\|\mathbf{w} - \mathbf{w}^*\|,\; \frac{2L}{\sigma_{\min}}\,\|\mathbf{w} - \mathbf{w}^*\|^2\right\}, \quad \text{where } \mathbf{w}^* = \operatorname*{argmin}_{\mathbf{w}}\mathcal{L}(\mathbf{w}),$$

*where $\kappa$ and $\sigma_{\min}$ are the conditioning number and smallest eiganvalue of $\nabla^2\mathcal{L}(\mathbf{w})$, respectively.*

**Proof** The lemma essentially follows via the standard analysis of the Newton's method. For the sake of completeness we will outline the proof following [WRXM17]. Denoting $\mathbf{H} = \nabla^2\mathcal{L}(\mathbf{w})$ and $\mathbf{g} = \nabla\mathcal{L}(\mathbf{w})$, we define the auxiliary function

$$\phi(\mathbf{p}) \stackrel{\text{def}}{=} \mathbf{p}^\top\mathbf{H}\mathbf{p} - 2\mathbf{p}^\top\mathbf{g}$$

By definition of $\phi(\mathbf{p})$ we have $\phi(\mathbf{p}^*) = \phi(\mathbf{H}^{-1}\mathbf{g}) = -\|\mathbf{p}^*\|_{\mathbf{H}}^2$. If follows that

$$\begin{aligned}
\phi(\widehat{\mathbf{p}}) - \phi(\mathbf{p}^*) &= \|\mathbf{H}^{\frac{1}{2}}\widehat{\mathbf{p}}\|^2 - 2\mathbf{g}^\top\mathbf{H}^{-1}\mathbf{H}\widehat{\mathbf{p}} + \|\mathbf{H}^{\frac{1}{2}}\mathbf{p}^*\|^2 \\
&= \|\mathbf{H}^{\frac{1}{2}}(\widehat{\mathbf{p}} - \mathbf{p}^*)\|^2 = \left\|\widehat{\mathbf{p}} - \mathbf{p}^*\right\|_{\mathbf{H}}^2 \leq \alpha^2\,\|\mathbf{p}^*\|_{\mathbf{H}}^2 = -\alpha^2\phi(\mathbf{p}^*).
\end{aligned}$$

We invoke the classical result in local convergence analysis of Newton's method [NW06], using the statement of Lemma 9 in [WRXM17].

**Lemma 15 ([WRXM17])** *Assume Hessian is L-Lipschitz and that $\widehat{\mathbf{p}}$ satisfies $\phi(\widehat{\mathbf{p}}) \leq (1 - \alpha^2)\min_{\mathbf{p}}\phi(\mathbf{p})$. Then $\widetilde{\mathbf{w}} = \mathbf{w} - \widehat{\mathbf{p}}$ satisfies*

$$\|\widetilde{\mathbf{w}} - \mathbf{w}^*\|_{\mathbf{H}}^2 \leq L\,\|\mathbf{w} - \mathbf{w}^*\|^2\|\widetilde{\mathbf{w}} - \mathbf{w}^*\| + \frac{\alpha^2}{1 - \alpha^2}\|\mathbf{w} - \mathbf{w}^*\|_{\mathbf{H}}^2.$$

Lemma 15 immediately implies that one of the following two inequalities hold:

$$\|\widetilde{\mathbf{w}} - \mathbf{w}^*\| \leq \frac{2L}{\sigma_{\min}(\mathbf{H})} \cdot \|\mathbf{w} - \mathbf{w}^*\|^2,$$

$$\|\widetilde{\mathbf{w}} - \mathbf{w}^*\| \leq \frac{\alpha}{\sqrt{1 - \alpha^2}}\sqrt{\frac{2\lambda_{\max}(\mathbf{H})}{\lambda_{\min}(\mathbf{H})}} \cdot \|\mathbf{w} - \mathbf{w}^*\|,$$

which proves Lemma 14. ∎

Note that Corollary 6 follows immediately by combining Corollary 4 with Lemma 14.

Figure 2: Comparison of the estimation error between *determinantal* and *uniform* averaging on four libsvm datasets.

## C Experiments

In this section, we experimentally evaluate the estimation error of determinantal averaging for the Newton's method (following the setup of Section 1.1), and we compare it against uniform averaging [WRXM17]. We use square loss $\ell_i(\mathbf{w}^\top \mathbf{x}_i) = (\mathbf{w}^\top \mathbf{x}_i - y_i)^2$, where $y_i$ are the real-valued labels for a regression problem, and we run the experiments on several benchmark regression datasets from the libsvm repository [CL11]. In this setting, the local Newton estimate computed from the starting vector $\mathbf{w} = \mathbf{0}$ is given by:

$$\widehat{\mathbf{p}} = \left( \frac{1}{k} \sum_{i=1}^n b_i \mathbf{x}_i \mathbf{x}_i^\top + \lambda \mathbf{I} \right)^{-1} \frac{1}{n} \sum_{i=1}^n y_i \mathbf{x}_i, \quad \text{where} \quad b_i \sim \text{Bernoulli}(k/n).$$

In all of our experiments we set the regularization parameter to $\lambda = \frac{1}{n}$. Let $\widehat{\mathbf{p}}_1, \ldots, \widehat{\mathbf{p}}_m \overset{\text{i.i.d.}}{\sim} \widehat{\mathbf{p}}$ be $m$ distributed local estimates and denote $\widehat{\mathbf{H}}_t$ as the $t$th local Hessian estimate. The two averaging strategies we compare are:

$$\text{determinantal:} \quad \widehat{\mathbf{p}}_{\text{det}} = \frac{\sum_{t=1}^m \det(\widehat{\mathbf{H}}_t) \, \widehat{\mathbf{p}}_t}{\sum_{t=1}^m \det(\widehat{\mathbf{H}}_t)}, \qquad \text{uniform:} \quad \widehat{\mathbf{p}}_{\text{uni}} = \frac{1}{m} \sum_{t=1}^m \widehat{\mathbf{p}}_t.$$

Figure 2 plots the estimation errors $\|\widehat{\mathbf{p}}_{\text{det}} - \mathbf{p}^*\|$ and $\|\widehat{\mathbf{p}}_{\text{uni}} - \mathbf{p}^*\|$, where $\mathbf{p}^*$ is the exact Newton step starting from $\mathbf{w} = \mathbf{0}$, for datasets ABALONE, CPUSMALL, MG[3] and CADATA [CL11] (for convenience, the plot from Figure 1 in Section 1.1 is repeated here). The reported results are averaged over 100 trials, with shading representing standard error. We consistently observe that for a small number of machines $m$ both methods effectively reduce the estimation error, however after a certain point uniform averaging converges to a biased estimate and the estimation error flattens out. On the other hand, determinantal averaging continues to converge to the optimum as the number of machines keeps growing. We remark that for some datasets determinantal averaging exhibits larger variance than uniform averaging, especially when local sample size is small. Reducing that variance, for example through some form of additional regularization, is a new direction for future work.

## Footnotes

[2]In [GCT12] it is assumed that $d \ge 3$, however this assumption is not used anywhere in the proof.

[3]We expanded features to all degree 2 monomials, and removed redundant ones.