[Reviews · NeurIPS 2019]

Reviewer 1



Quality & Clarity. This paper is well-written. Te exposition throughout the text is clear and appropriated. Section 2 is particularly neat! Originality: The paper tackles a central problem in distributed optimization and numerical linear algebra: estimating the inverse Hessian matrix from local machines. This paper shows an elegant solution (a weighted average) to solve the bias problem faced when one is estimating the inverse Hessian using a simple average of the local estimators. Significance: The two discussed applications are very relevant to ML, Newton’s method and uncertainty quantification. In fact, the sole contribution to distributed Newton's method would probably be of significant impact to the community; and the method could also generalize to other settings/applications. ---- I acknowledge that I have read the author's response, as well as the other reviewers. Thanks for the response: I have read the paper again and changed my original review and score.

Reviewer 2



Additional Comments: • Overall, the article is well-written and structured. It has a clear contribution and also significant theoretical justification. There are only a few mistyping and grammatical errors: Line 17: “tranformation” “transformation” Line 133: “ its entries is of …” “its entries are of …” Line 146: “ accross” “across” Line 150: “ emprical” “empirical” • In line 54, it is better to define in the theorem 2 along with . • In line 64, it seems the upper bound of the summation operator is incorrect. • Although plotting the estimation error in figure 1 is a useful tool to justify the superiority of the proposed method, using different markers or colors makes the graph more readable. • Line 92 and 104: sections are referenced incorrectly. • In line 93, I think should be since it is assumed that there are estimators unless the purpose is its approach to infinity to reduce the estimation error. If there is no constraint on the number of estimators, this means choosing subsamples is with replacement. However, in line 158, the authors claim that their method is without replacement. • In line 110, it deserves that Sylvester’s theorem is referenced. • To be more illustrative, for each part of the paper separately, provide your assumptions for variables clearly. For example, in line 183, specifying as “independently random variable” is more illustrative than “independently random”. • It seems the proof of the lemma 7 is not totally correct. In lemma 7, the authors are trying to prove two statements of “a” and “b” using inductive reasoning. However, in the line corresponding to the inductive hypothesis, they use the statement “b” to prove the statement “a”. It is more acceptable to prove the statement “b” beforehand and then use it as an axiom for proving “a”, or use assumptions and axioms which hold for both statements to prove them simultaneously. • In line 206, defining is required to determine the bounds for the variable “x” in Lemma 8. • The proof for the Lemma 9 is somewhat confusing. For example, it is difficult to find the connection between the interval of “z” in line 224 and the bound of “x” in Lemma 8. Besides, it seems by considering “” as a Bernouli random variable with the mean of “” and “” as a constant, the equation in line 223 is no longer correct. • There are similar works in the literature that attempt to improve the approximation of the inverse Hessian such as [1], [2], and [3]. I would recommend that the authors cite related works with mutual concerns to motivate their paper more. Moreover, I suggest that the authors provide evidence of the advantages of their proposed approach with these related works in terms of performance and complexity in details. References: 1. Gower, R., Hanzely, F., Richtárik, P. and Stich, S.U., 2018. Accelerated stochastic matrix inversion: general theory and speeding up BFGS rules for faster second-order optimization. In Advances in Neural Information Processing Systems (pp. 1619-1629). 2. Mokhtari, A., Ling, Q. and Ribeiro, A., 2016. Network Newton distributed optimization methods. IEEE Transactions on Signal Processing, 65(1), pp.146-161. 3. Bajovic, D., Jakovetic, D., Krejic, N. and Jerinkic, N.K., 2017. Newton-like method with diagonal correction for distributed optimization. SIAM Journal on Optimization, 27(2), pp.1171-1203.

Reviewer 3



Originality: This work builds upon previous work but derives several results that are interesting in their own right, including moment bounds for determinants and adjugates of random matrices. Quality & Clarity: This paper is extremely clear and easy to read, which contributes significantly to its quality. The results are carefully proven and explained, and the proofs are easy to follow. Significance: the ability to recover the inverse of a large matrix through distributed averaging and without bias has many important applications to machine learning problems, as the authors point out. ----- Questions ------ - Would it be possible in certain of the ML scenarios mentioned, to take advantage of the determinantal weights by sampling subsets of size k that yield high determinantal weights? - Do you know if the identities derived in Section 2 can be generalized to other elementary symmetric polynomials?

[Author Response · NeurIPS 2019]

## To Reviewer 1

Thanks for the feedback! We would like to stress that our main contribution is a *novel theoretical result* (Theorems 1 and 2) that is of considerable interest to the machine learning community. For this reason, we deliberately structured the paper to highlight the technical analysis in the main body. We believe that proposing a new method which - for the first time - *completely eliminates inversion bias from distributed estimation* is significant enough to stand on its own. However, we do provide numerical experiments plotting the estimation error on four benchmark datasets, all of which clearly support our analysis. One of the plots is shown in Section 1.1, while the remaining plots and a detailed discussion of the experiments is shown in Appendix C. We completely agree that further empirical evaluation is needed, for example incorporating determinantal averaging into several different distributed second-order optimization methods (such as GIANT [WRKXM18] and DANE [SSZ14]) and comparing the effects, however this is beyond the scope of this work. For the final version of this paper, we will be happy to provide a conclusion section at the end, where we will emphasize that our main result holds very generally and is not just applicable to the Newton's method, the main motivation here, but also to other linear functions of inverse Hessians as well as inverse covariance matrices which are of interest in uncertainty quantification.

## To Reviewer 3

Thanks for all the comments and suggestions! We will be sure to incorporate them into the final version. Also, we will definitely mention the extra references.

- Regarding line 158 and with/wihout replacement sampling, what we meant is that the sample we use for a single estimator is drawn without replacement. However, as you observed, when there is multiple estimators, the same point may be used in several of them.

- Regarding the proof of Lemma 7, the proof structure is the following: the inductive hypothesis for size $n$ consists of both statements (a) and (b). We then prove statement (a) for size $n + 1$ using the inductive hypothesis (employing both (a) and (b) for size $n$). Finally, we prove statement (b) for size $n + 1$ by using (already proven) statement (a) for size $n + 1$.

- Thanks for the feedback regarding the proof of Lemma 9. We will clarify it in the final version.

## To Reviewer 4

Thanks for the comments and questions! We will address the comments in the final version. We answer your questions below:

- There are methods which sample estimators with probability proportional to certain determinantal weights (those weights are similar to ours, but not identical), such as volume sampling [DW17] (see lines 163-167 in the paper). However, unavoidably those sampling techniques are so computationally expensive that this cost alone makes them impractical in most distributed settings.

- We do not know whether the identities from Section 2 extend to elementary symmetric polynomials, however this is quite possible and it is a very interesting question indeed!

## References

[DW17] Michał Dereziński and Manfred K. Warmuth. Unbiased estimates for linear regression via volume sampling. In *Advances in Neural Information Processing Systems 30*, pages 3087–3096, Long Beach, CA, USA, December 2017.

[SSZ14] Ohad Shamir, Nati Srebro, and Tong Zhang. Communication-efficient distributed optimization using an approximate Newton-type method. In Eric P. Xing and Tony Jebara, editors, *Proceedings of the 31st International Conference on Machine Learning*, volume 32 of *Proceedings of Machine Learning Research*, pages 1000–1008, Bejing, China, 22–24 Jun 2014. PMLR.

[WRKXM18] Shusen Wang, Farbod Roosta-Khorasani, Peng Xu, and Michael W Mahoney. Giant: Globally improved approximate newton method for distributed optimization. In S. Bengio, H. Wallach, H. Larochelle, K. Grauman, N. Cesa-Bianchi, and R. Garnett, editors, *Advances in Neural Information Processing Systems 31*, pages 2332–2342. Curran Associates, Inc., 2018.


[Meta-Review · NeurIPS 2019]

There were disagreement in the opinion regarding this paper. However, the rebuttal did manage to convince reviewers in the positive direction. After the discussion, there is now a consensus among reviewers that this paper is well above the acceptance threshold. Authors are encouraged to take reviews into account before preparing the final version.